# Improving continence management for people with dementia in the community in Aotearoa, New Zealand: Protocol for a mixed methods study

**Vanessa Burholt**[1,2,3]*, **Kathryn Peri**[1], **Sharon Awatere**[1], **Deborah Balmer**[1], **Gary Cheung**[4], **Julie Daltrey**[1], **Jaime Fearn**[5], **Rosemary Gibson**[5], **Ngaire Kerse**[2], **Anna Michele Lawrence**[6], **Tess Moeke-Maxwell**[1], **Erica Munro**[1], **Yasmin Orton**[1], **Avinesh Pillai**[7], **Arapera Riki**[1], **Lisa Ann Williams**[1]

1 School of Nursing, Faculty of Medical and Health Sciences, University of Auckland, Auckland, New Zealand, 2 School of Population Health, Faculty of Medical and Health Sciences, University of Auckland, Auckland, New Zealand, 3 Centre for Innovative Ageing, College of Human and Health Sciences, Swansea University, Wales, United Kingdom, 4 Department of Psychological Medicine, Faculty of Medical and Health Sciences, University of Auckland, Auckland, New Zealand, 5 School of Psychology, Massey University, Palmerston North, New Zealand, 6 Department of Urology, Counties Manukau, Te Whatu Ora, Auckland, New Zealand, 7 Department of Statistics, Faculty of Science, University of Auckland, Auckland, New Zealand

☯ These authors contributed equally to this work.
* Vanessa.burholt@auckland.ac.nz

**Data Availability Statement:** The datasets generated and/or analysed during the study will not be made publicly available as restrictions apply to

## Abstract

### Background

The number of people living with dementia (PLWD) in Aotearoa New Zealand (NZ) was estimated at 96,713 in 2020 and it is anticipated that this number will increase to 167,483 by 2050, including an estimated 12,039 Māori (indigenous people of NZ) with dementia. Experiencing urinary incontinence (UI) or faecal incontinence (FI) is common for PLWD, particularly at the later stages of the disease. However, there is no robust estimate for either prevalence or incidence of UI or FI for PLWD in NZ. Although caregivers rate independent toilet use as the most important activity of daily living to be preserved, continence care for PLWD in the community is currently not systematised and there is no structured care pathway. The evidence to guide continence practice is limited, and more needs to be known about caregiving and promoting continence and managing incontinence for PLWD in the community. This project will seek to understand the extent of the challenge and current practices of health professionals, PLWD, caregivers and family; identify promising strategies; co-develop culturally appropriate guidelines and support materials to improve outcomes; and identify appropriate quality indicators so that good continence care can be measured in future interventions.

### Methods and analysis

A four-phase mixed methods study will be delivered over three years: three phases will run concurrently, followed by a fourth transformative sequential phase. Phase 1 will identify the

the availability of these data (intention of data analysis included in participant information forms) and sensitivity (i.e. human data, Māori data sovereignty). These restrictions have been ratified by the Auckland Health Research Ethics Committee (AHREC) at the University of Auckland and the Southland Health and Disability Ethics Committee. Data will be available from AHREC (ahrec@auckland.ac.nz) on reasonable request. Data will be located in a controlled access repository at the University of Auckland.

**Funding:** This work was supported by Health Research Council of New Zealand (https://www.hrc.govt.nz/) Project Grant 21/117 to Vanessa Burholt. The funders had and will not have a role in study design, data collection and analysis, decision to publish, or preparation of the manuscript.

**Competing interests:** The authors have declared that no competing interests exist.

prevalence and incidence of incontinence for PLWD in the community using a cohort study from standardised home care interRAI assessments. Phase 2 will explore continence management for PLWD in the community through a review of clinical policies and guidance from publicly funded continence services, and qualitative focus group interviews with health professionals. Phase 3 will explore experiences, strategies, impact and consequences of promoting continence and managing incontinence for PLWD in the community through secondary data analysis of an existing carers' study, and collecting new cross-sectional and longitudinal qualitative data from Māori and non-Māori PLWD and their caregivers. In Phase 4, two adapted 3-stage Delphi processes will be used to co-produce clinical guidelines and a core outcome set, while a series of workshops will be used to co-produce caregiver resources.

## Introduction

### The scale of the issue

The number of people living with dementia (PLWD) in Aotearoa New Zealand (NZ) was estimated at 96,713 in 2020 and it is anticipated that this number will increase to 167,483 by 2050, including an estimated 12,039 Māori (indigenous people of NZ) with dementia [1]. Māori with dementia presenting to hospital-based memory services are significantly younger than their NZ European peers [2]. It is estimated that there are more than 50 million people living with dementia worldwide, with the number expected to reach 152 million in 2050 [3].

Experiencing urinary incontinence (UI) or faecal incontinence (FI) is common for PLWD particularly at the later stages of dementia [4]. Urinary incontinence is defined as the involuntary loss of urine, and faecal incontinence is the involuntary loss of solid or liquid faeces [5].

Dementia also has an impact on the physiological functions of the body and can contribute to bladder or bowel dysfunction [5]. However, incontinence is more frequently the result of cognitive impairment [5]. Cognitive decline associated with all types of dementia can interfere with activities of daily living such as toilet use. For example, with moderate or severe dementia, older people may experience difficulties decoding sensations associated with the need to void, or inaccurately estimating how long they require to get to and use the toilet. The latter may be compounded by struggles recalling the whereabouts of a toilet [6].

There is limited and widely varying data on the prevalence of UI and FI in the population of PLWD in the community. Studies estimate that prevalence of UI (in high income countries) is in the range of 10–84% [7, 8] and the prevalence of FI ranges between 0.9–27% [9]. Studies that combine UI and FI estimate prevalence at 34% for PLWD in the community [10, 11]. To date, there is no robust estimate for either prevalence or incidence of UI and FI for PLWD in NZ. Furthermore, there is no robust research on the prevalence of incontinence for older Māori living with dementia, despite a higher prevalence of incontinence in adult Māori than non-Māori populations [12, 13].

### Continence services and support

Dementia impacts on the quality of life of the people living with the disease, their family and caregivers. Research has shown that dementia significantly contributes to both to disability and the need for care among older adults [14]. Caregivers supporting PLWD consider the preservation of independent toilet use as more important than autonomy for other activities of

daily living [15]. However, a review of literature published in English from countries in the upper quartile (above 47) of the Human Development Index demonstrated that continence care in the community is currently not systematised for PLWD and there is no structured care pathway for this population [16].

Promoting continence or managing incontinence in the community for PLWD is dependent on timely diagnosis, assessment (including addressing contributory causes such as infection, pharmacological side effects and reduced mobility), regular review, generalist and specialist intervention [16, 17]. Assessments and access to services, support and continence products, are mainly administered through primary care [18]. However, PLWD and/or caregivers may be reluctant to raise continence issues with primary care health professionals as they may be embarrassed talking about the stigmatised subject, wish to protect their privacy or worry that revealing such problems may lead to a residential care placement for the PLWD [6, 18–20]. Therefore, it is important for clinicians and health professionals to initiate conversations and plan support so that caregivers are not left to cope alone [6]. However, health professionals may be reluctant to initiate conversations about continence issues with PLWD as there is a lack of specific continence guidance addressing dementia [16, 21, 22] and insufficient evidence to support nonpharmacological and non-surgical conservative interventions, environmental adaptations or behavioural management (e.g., prompted/timed voiding) [5, 21, 23].

Promoting continence and managing incontinence for PLWD in the community often requires hyper-vigilance by caregivers. This, in turn can contribute to sleeplessness and exhaustion for the caregiver, increasing the likelihood of residential care placement for the PLWD [15, 20, 24, 25]. Presently, we do not know how unpaid caregivers, and families in NZ manage the challenges associated with promoting continence and managing incontinence.

Some caregivers and PLWD face additional challenges: socio-cultural position, geographic location or area deprivation [26] can influence inequitable access to health, social care and community services [27, 28]. Rates of moving into residential care are much greater for PLWD than for older people without dementia, and incontinence is frequently identified as a predictor for institutionalisation in this population [29]. This suggests that more can be done to support caregivers and family in the community.

There is evidence of inequities in access to health services in NZ for Māori. For many Māori, whānau (family) care is often perceived as 'normal' [30–32] and is often cited as a reason why unpaid caregivers do not seek out services [33]. However, research has indicated that in addition to the macro-level structural issues noted above, colonialisation [34], discrimination in health care services [35], bureaucratic obstacles [36], and difficulties navigating health and social care systems [37–39] can also create barriers to accessing health care services for many unpaid caregivers and care-recipients. Mahi aroha (care work) informed by Māori cultural concepts such as aroha (love and compassion), manaakitanga (care), whakapapa (genealogy) and whanaungatanga (relationships/ connections) is especially crucial in promoting hauora (Māori philosophy of health) and creating an environment that supports wairua (spiritual wellbeing) [31, 40]. Health services and practices that resonate with the diverse cultural values and contexts of Māori, and that include appropriate outreach mechanisms are vitally important [41–43]. Inadequate cultural competency and safety within the health system [44, 45] is likely to negatively impact on subsequent support-seeking behaviours in relation to continence services.

## Summary

There is very little evidence of a structured care pathway to promote continence or manage incontinence for PLWD in the community. Robust research is required to examine and

establish the prevalence and incidence of UI and FI for PLWD in the community so that we can estimate service or intervention need. The evidence to guide practice is limited, and more needs to be known about the strategies that PLWD and unpaid caregivers use in the community [16]. Prior to developing a body of evidence establishing the effectiveness of services and interventions to promote continence or manage incontinence, we need to cultivate our understanding of 'good outcomes' from the perspective of health professionals, PLWD, unpaid caregivers and families [16].

This 3-year project is the first stage in a pipeline of research in which we seek to understand the extent of the challenge, and current practices of health professionals, PLWD, and unpaid caregivers. We will identify promising strategies for promoting continence and managing incontinence for PLWD in the community and co-develop culturally safe guidelines and support materials intended to improve outcomes. We will identify appropriate quality indicators for PLWD, their unpaid caregivers and family so that good continence care can be measured in future complex interventions.

## Objectives

The objectives of this study are to:

- Establish the prevalence and incidence of UI and FI among PLWD in the community who have had an interRAI home care assessment

- Collate clinical guidance for continence management, and hold workshops with clinicians and health care professionals, in order to understand continence policy and practice in relation to PLWD in the community, their caregivers and family

- Undertake interviews with PLWD, their caregivers and family in order to understand experiences, strategies, impacts and consequences of promoting continence and managing incontinence in the community

- Develop tools and resources to support researchers, clinicians, health care professionals, PLWD, their caregivers, and family to promote continence and manage incontinence in the community.

## Research questions

The study addresses the following four overarching research questions:

1. What is the prevalence and incidence of UI and FI among PLWD in the community who have had an interRAI home care assessment? (Phase 1)

2. Does the clinical guidance for continence management provided by specialist services and current practice address issues experienced by PLWD in the community, their caregivers and family? (Phase 2)

3. What are the experiences, strategies, impacts and consequences of promoting continence and managing incontinence from the perspective of (a) PLWD, (b) caregivers and family; and how do these differ between geographic locations and ethnicity, and change over time? (Phase 3)

4. What tools would support self/caregiver, and family/practitioner management of continence for PLWD and how should effectiveness be evaluated? (Phase 4)

## Methods

The theoretical framework has been developed in preliminary work with an expert group including PLWD, caregivers and practitioners [16]. The theoretical framework considers continence and dementia across the scientific spectrum. We do not address neuropathology of dementia and incontinence. However, this study examines the extent of the challenge, gateways to continence support (policies, guidelines and assessment), interventions or strategies for promoting continence or managing incontinence and the impact of these elements on psychological, social, physical, material and environmental outcomes for PLWD, caregivers and families. The model also takes into account dynamic inter-relationships between domains, and the influence of personal resources (material resources, education and health literacy, and social support), socio-cultural factors (e.g., norms, values and beliefs) and environmental contexts (e.g., rural or urban location and area disadvantage) on availability of interventions and psychosocial outcomes.

Drawing on a framework for developing complex interventions [46] the study represents the first steps in the development or identification of an intervention. The 3-year project is the first stage in a pipeline of research, and considers context (Phase 1–3), programme theory (Phase 2), diverse stakeholders' perspectives, and key uncertainties (Phases 2–4) to identify the most promising strategies and interventions for promoting continence and managing continence for PLWD living in the community (Phase 4) for future testing [46].

The study draws on a pre-tested rigorous methodology [47]. The first three phases of the research will run concurrently, followed by a fourth transformative sequential phase (Fig 1). Phases 2–4 utilise a participatory action research approach [48, 49], often referred to as integrated knowledge translation. This approach ensures the ongoing relationship between researchers and community groups for the purpose of engaging in mutually beneficial research to support decision-making [50–52].

### Phase 1: Prevalence and incidence of incontinence for PLWD in the community

**Aims.** To investigate,

i.  The prevalence of UI and FI in community-dwelling older people with dementia;

ii.  The incidence of UI and FI in community-dwelling older people with (dementia;

iii.  The risk of UI and FI for older people with dementia by age, gender, ethnicity, geographic location, and co-morbidity.

**Sample and setting.** interRAI-HC is a standardised comprehensive geriatric assessment with ∼250 clinical and psychosocial variables. All community-dwelling New Zealanders receiving publicly funded home support and/or personal care are required to be assessed by the interRAI home Care (interRAI-HC) assessment. Trained assessors conduct home care assessments that are repeated every 6 months for people deemed eligible for services. Assessments occur more frequently if there is a change in need for support. Although the sample of older people receiving home care assessments is not representative of the general population, this is a pragmatic utilisation of routinely collected data to rapidly map community epidemiology. Furthermore, it is likely more robust than a majority of previous epidemiological studies based on small convenience samples of participants [16].

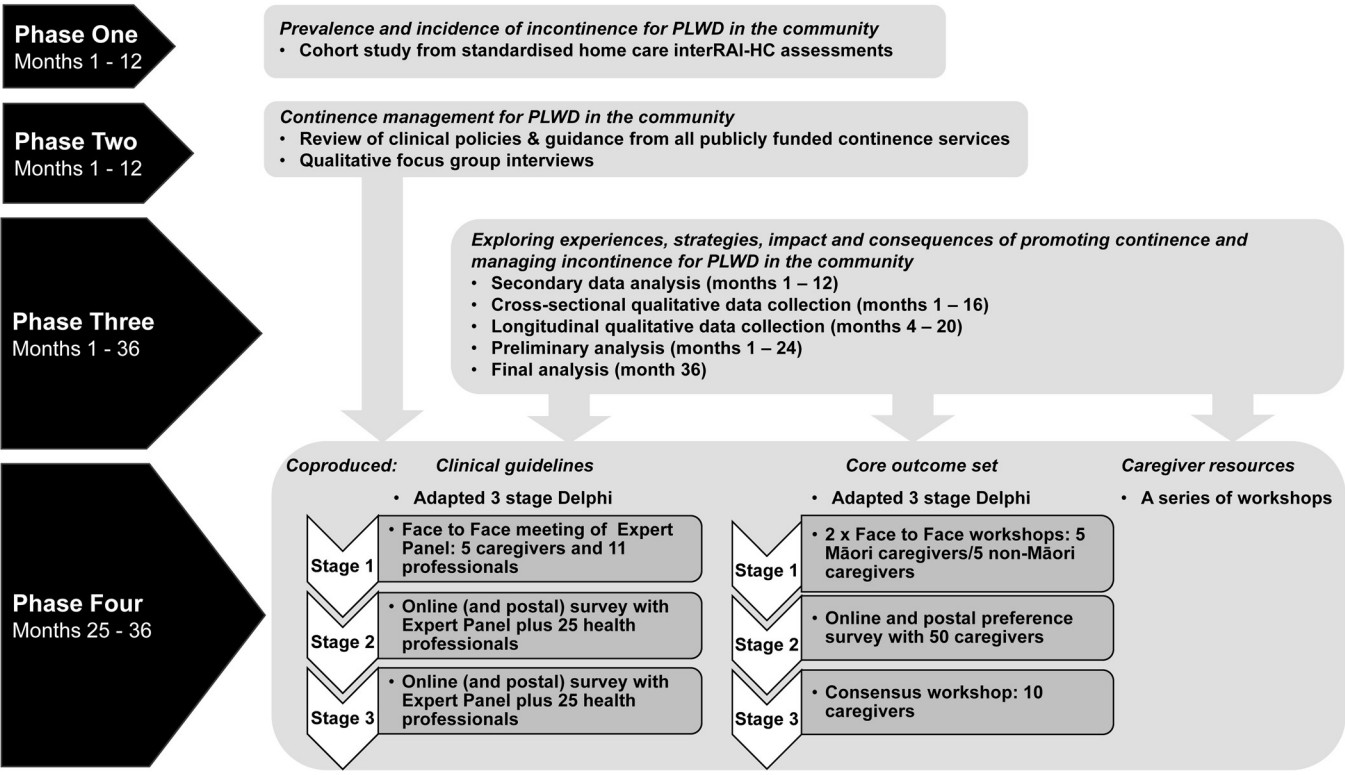

**Fig 1. Phases and activities of the study.**

Our sample comprises all older people (non-Māori 65+ years; Māori 55+ years), who had an interRAI-HC assessment in a 5-year period from 1 Aug 2016 to 31 July 2021. Data from 1 Aug 2020 to 31 July 2021 will be used to establish 12-month period prevalence of UI and FI for older Māori and non-Māori 65+ years with a diagnosis of dementia. interRAI-HC assessment routinely collects information on the diagnoses of "Alzheimer's disease" and "Dementia other than Alzheimer's disease". These two diagnoses will be used to identify study participants with a diagnosis of dementia [53]. Our primary outcomes (UI and FI) are also routinely enquired as part of an interRAI-HC assessment.

For the incidence analysis, participants in the dementia cohort will be followed up from the day of the first dementia diagnosis during the period 1 Aug 2016 and 31 July 2021. The non-dementia cohort comprised all eligible older people without a dementia diagnosis during the 5-year period and will be followed up from the first assessment during the 5-year study period. For both cohorts, in separate analysis participants with prevalent UI or prevalent FI (i.e., UI or FI recorded on the first assessment during the 5-year period) and those with no follow-up assessment will be excluded from the analysis. All participants will exit the study on the date of their last assessment or 31 July 2021.

**Study design.**    A cohort study of routinely collected data from standardised interRAI-HC assessments.

**Data analysis.**    Incidence of FI or UI will be established in separate analysis for each form of incontinence, and for each cohort (dementia cohort, or non-dementia cohort) by dividing the number of new cases of incontinence by the sum of months in the study and expressed as the rate per 100 person years.

Logistic regression will be used to identify independent risk factors that predict the onset of FI or UI, including socio-demographics, (age [54–56], gender [54, 55], ethnicity [57]) chronic disease status (depression [57], stroke [58], diabetes [54, 58], Parkinson's disease [56]), lifestyle factors (physical exercise [56, 59], smoking [55, 56]) functional variables (independence in activities of daily living [55, 56]) and environmental characteristics (access to rooms in the home [60], area deprivation [56], and rural/urban classification [57]). Survival curves of the populations will be compared with Cox regression analysis adjusted for the variable that are significantly related to incident UI or incident FI. Differences in the incidence of FI and UI between the dementia cohort and non-dementia cohort will be expressed with the hazard ratio.

## Phase 2: Continence management for PLWD in the community

**Aims.** To investigate,

i. The extent to which clinical guidance for incontinence management provided by specialist continence services and primary care providers addresses issues for PLWD in the community, their caregivers and family; and

ii. Primary care, community health professionals and voluntary staff perceptions of effective strategies for promoting continence and managing incontinence for PLWD in the community.

**Study design.** A review of clinical policies and guidance collated from all regional publicly funded continence services. Local clinical guidelines for incontinence provide clinical practice direction for primary health care professionals supporting older PLWD in the community.

Qualitative group interviews with a purposive sample of ≈80 people in 7–8 focus groups, each comprising a discrete set of health care professionals working in NZ (e.g. General Practitioners, Nurse Practitioners, Continence Nurses, dementia support workers, physiotherapists), will be asked about the solutions that would be offered to promote continence or manage incontinence in response to four vignettes. Discussions will cover views on strategies, advice, interventions, aids, support or technology, perceptions of gaps in the knowledge and/or provision to support people and their unpaid caregivers and perceptions of effective strategies (S1 Appendix, Focus Group Questions).

**Analysis.** Interpretative policy analysis [61] comprising documentary analysis of international guidelines [62] and regional publicly funded continence services policy aspirations and an analysis of staff perceptions and experiences. Shifting focus away from instrumental rationality (e.g., 'productivist' approaches such as 'active ageing' [63])), to value rationality in interpretation and implementation of public policies, qualitative interpretive policy analysis asks 'what are the meanings of a policy?' [61, 64, 65]. Analysis of focus group discussions will examine symbolic artefacts of language (e.g., used to describe PLWD and continence challenges), objects (e.g., containment products) and acts, that are significant carriers of policy meanings [60]. Analysis will identify 'interpretive communities' or shared group understandings of policy ideals and language [61, 64, 66].

## Phase 3: Exploring experiences, strategies, impact and consequences of promoting continence and managing incontinence for PLWD in the community

**Aims.**   To investigate experiences, strategies, impact and consequences of promoting continence and managing incontinence from the perspective of (i) PLWD; (ii) caregivers and family; and changes over time, in different settings, and for Māori and non- Māori.

**Samples and settings.**   Extant qualitative data (open ended questions from a nationwide survey) conducted via regional dementia support service in 2016–17 for 526 caregivers to PLWD on sleep and wellbeing [24], plus follow-up interview data with 20 caregivers for whom the PLWD had recently transitioned into residential care [25]. Free text responses to the survey and interview data highlighting factors affecting sleep and wellbeing including continence identified in 94 of the 526 surveys and 13 of the 20 interviews.

A new cross-sectional convenience sample of PLWD (N≈30; with capacity to consent and experiencing incontinence issues), and their caregivers and family (N≈75): self-selected volunteers in response to approaches from collaborating community group networks, service user groups, and free local media (radio, newsletters) advertising the study across Hawkes Bay, Waikato, Auckland and Northland regions of North Island, NZ.

A new longitudinal convenience sample of caregivers and PLWD: self-selected volunteers with capacity to consent (5–10 Māori and 5–10 non-Māori care recipient-giver dyads or care recipient and family care networks) and invited to participate from the new cross-sectional sample.

**Study design.**   Secondary research (see data analysis) of extant qualitative data on caregivers of PLWD in relation to sleeplessness and the impact on caregiving across the trajectory of dementia to identify spontaneous and unprompted references to (in)continence (a common theme outside the scope of the original study).

A new cross-sectional qualitative study over a 16-month period. PLWD, caregivers, and family would be invited to take part in face-to-face guided interviews. Subsequently, PLWD and their caregivers or family would be invited to take part in Waves 2–4: a longitudinal qualitative follow-up study comprising serial qualitative face-to-face guided conversations every 4 months (with a maximum of 3 interviews) until the end of data collection period (end of month 20). Participants may take part in as many or as few interviews as they choose. In Waves 2–4, if a PLWD is no longer able to provide informed consent, only consenting caregivers will continue to be interviewed. For PLWD who leave the study because they have died, or entered residential care, caregivers will be invited to participate in one more interview only.

Māori interviews will be informed by culturally safe research processes. Recruitment and interviews will be informed by a pōwhiri model of engagement to ensure each participant is cared for in a culturally meaningful reciprocal knowledge exchange. This process upholds rangatiratanga (the right of Māori people to rule themselves; self-determination), whakapapa (genealogical or ancestral lineages; interconnections between Māori), mana (status), tikanga (cultural practices) of Māori based on the principles of whanaungatanga (relationship building) and manaakitanga (nurturing, care and hospitality) [67].

Surveys will incorporate validated instruments and data will be used to construct participant profiles characterising the samples of PLWD and their caregivers [47]. For PLWD: Mini Addenbrooke's Cognitive Examination [68]. For caregivers: health-related quality of life (European Quality of Life-5 Dimensions: EQ–5D 3L [69] and Carers of Older People in Europe (COPE) Index [70].

Sections in the semi-structured interview schedules will address toilet use, nocturia, UI and FI. The guide for each section will cover a description of the challenge(s); the effectiveness of

strategies adopted to address the challenge(s) (e.g., routines and practices, use of continence products, food, drink, medications, alternative or complementary therapies and rongoa (traditional Māori healing)); experience of discussing challenges with health professionals; and impact on everyday life (S2 Appendix, Topic guide PLWD cross-sectional interview; S3 Appendix, Topic guide caregiver cross-sectional interviews; S4 Appendix, Topic guide PLWD longitudinal interview; S5 Appendix, Topic guide caregiver longitudinal interview). Māori versions of the topic guides will substitute common Te Reo Māori words for English where appropriate (e.g., wharepaku instead of toilet) and will be developed by the Māori Research Fellow with support from a Māori Research Assistant (kuia, female Māori elder).

**Analysis.**   Survey comments (extant data) and transcripts will be anonymised [71], imported and coded using QSR NVivo software. Preliminary thematic analysis of secondary survey data and primary cross-sectional qualitative data [72], plus narrative analysis of the retrospective secondary interview data from caregivers [73], alongside interpretative phenomenological analysis of interview data from PLWD [74] will address the aims and results will feed into Phase 4.

A descriptive analysis of quantitative survey data will be used to construct participant profiles (e.g. changes in cognitive status, difficulties with toilet use, and incontinence over time) [47]. Longitudinal qualitative analysis will use time-ordered sequential framework analysis facilitating comparison over time within and between care recipient-caregiver or family relationships.

A Kāhui Kaumātua (advisory group) will help guide interpretation of Māori data analysis. Māori and non-Māori researchers will regularly discuss interpretations to understand converging/diverging needs and experiences.

## Phase 4: Co-produced clinical guidelines, a core outcome set (COS) and caregiver resources

**Aims.**   i.  To formulate and co-produce clinical/practice guidelines

ii.  To identify a core outcome set (COS) of quality indicators to be used to assess effectiveness in future complex interventions; and

iii.  To formulate, co-produce and test practical resources that support caregivers and family to promote continence and manage incontinence for PLWD in the community.

**Samples and settings.**   Health care professionals from Phase 2, and caregivers and family from Phase 3 who have indicated interest in Phase 4. The invitation to participate would also be extended to former caregivers to PLWD who have experienced challenges associated with promoting continence and managing incontinence in the community. Former caregivers will be self-selected volunteers in response to approaches from collaborating community group networks, service user groups, and local media advertising the study across Hawkes Bay, Waikato, Auckland and Northland regions of North Island, NZ. For samples for each co-production activity, see Fig 1.

**Study design.**   Different 3-stage adapted Delphi consultations [47, 75] will be used to co-produce clinical/practice guidelines and COS. A series of workshops with Māori and non-Māori caregivers will be convened for ideation and development of appropriate resource materials/models to support PLWD, caregivers and family.

*Co-produced clinical/practice guidelines*. Results from preliminary analysis of Phases 2 and 3 data alongside expert opinion from specialist health care professionals and caregivers will ensure that "patient values guide all clinical decisions" [76, 77]. Although participants never

meet or interact directly in the classic Delphi study [47, 78], research methods will be explained to lay members in Stage 1 during a face-to-face meeting. This will be followed by two consultation rounds to establish consensus and face validity of the principles [79].

- Stage 1: Agreement on the range of principles to be incorporated into draft clinical/practice guidance on promoting continence and managing incontinence for PLWD in the community.

- Stage 2: Consultation (by online survey and paper survey) with a wider range of continence and dementia professionals. The survey will include Likert-type responses (scale 1–9) to indicate the importance of each principle, open-ended questions for participants to add comments, and an opportunity to add to the list of principles [80]. The data obtained will be subjected to descriptive statistical analyses. Indicators rated with an overall median validity of <6 will be discarded after this round.

- Stage 3: Results of analysis of Stage 2 will be incorporated into a second online (or postal) survey (i.e. median validity score for each item), some original items may be amended and others added (based on stage 2 feedback). Items receiving an overall median rating of 8–9 will be included in the final guidance.

*Co-produced core outcome set*. A Core Outcome Set (COS) is a list of critically important outcome domains that should be measured in relation to a specific disease or trial population [81, 82]. In an accessible modified Delphi approach, a minimum set of outcome domains will be identified, that should be measured and reported in pharmacological and non-pharmacological interventions or clinical trials relating to the promotion of continence and management of incontinence for PLWD in the community, conforming to the 11 minimum standards for COS developers [82]. Drawing on earlier phases of the study [83–85] PLWD will inform the design of the COS [86, 87].

- Stage 1: A list of candidate items would be generated, drawing on three sources: (i) outcomes identified in Phases 2 and 3, (ii) extant COS for PLWD (but not specific to continence issues) [88, 89] and (iii) extant COS for incontinence (but not specific to PLWD). During two workshops the candidate items will be refined (removing duplication, consolidating areas of commonality and mapping outcome items into domains) [88].

- Stage 2: The results of Stage 1 workshops would be combined, and each item would be framed as a survey question to ascertain the importance of outcomes using a three-point response [86]. The preference survey would be distributed to current and past Māori and non-Māori caregivers. Items will be identified as 'consensus included'; 'consensus excluded'; and 'no consensus' [86]. Split group analysis will identify any difference in outcome preferences between Māori and non-Māori respondents.

- Stage 3: In the final consensus workshop items for which there are between group differences for 'consensus included' and 'consensus excluded', along with those that were classified as 'no consensus' would be discussed and workshop participants would vote anonymously in real time using an audience response system [e.g. 90] to keep or omit items from the final list of outcomes. Maps of domains from Stage 1 will be used to group items, and between-workshop differences discussed until agreement on domain placement is reached.

Measurement of domains will not be defined [91], as this comprises one of the next steps in the pipeline of research.

*Co-production of resources*. A series of workshops will be organised to devise and co-create resources. Workgroup discussion would focus on prioritising the 'most promising' strategy/ies

identified in analysis of data in Phase 3, and how best to incorporate this/these into resource (s). Methods of co-production and the resources outputs will depend on the choices made during the workshops, but might include digital storytelling, documentary film-making, co-design of function and content of a digital platform/educational hub, or co-creation of informational leaflets. Co-produced resources would be piloted with a small group of Māori and non-Māori caregivers to gain feedback for iterative improvement.

**Ethics.**   Ethical approval for Phase 1 was obtained from the Auckland Health Research Ethics Committee (reference AH23238) on the 1 September 2021.

Ethical approval for Phase 2 was obtained from the Auckland Health Research Ethics Committee (reference AH23747) on the 10 October 2021. In Phase 2, health professionals will be provided with a participant information sheet and will be asked to provide written consent to participate in focus groups.

Ethics approval for Phase 3 was obtained from Southern Health and Disability Ethics Committee (reference 11658) on the 28 April 2022. After first contact with the research team (via email or phone as indicated on the advert), potential participants will be sent a copy of the Participant Information Sheet (PIS) by post or email. Separate PIS will be provided for caregivers and PLWD, and for cross-sectional and longitudinal interviews. The research team will follow-up after two weeks and make an appointment for interview if the potential participant wishes to take part in the study. At this time, participants will be asked if they have any more questions about the study that they would like answered. Participants will be asked to give written informed consent on the day of the interview, prior to the interview commencing. Judgments concerning capacity to provide informed consent will be made by trained researchers. The researchers will ask participants to describe in their own words what the study is about, and what they are being asked to do. Consent will be on-going throughout the study and participants will be asked to consent (with the relevant judgements of capacity made) at every data collection point.

Ethical approval for Phase 4 will be sought towards the end of Year 2 (e.g., September 2023). Participants in face-to-face focus groups will be asked to provide written consent, and participants completing written or online surveys will be asked to indicate consent by agreeing to a privacy and consent section before submitting the survey.

## Results

Fig 1 outlines the timing of the study. Phases 1–3 commenced in October 2021. Phase 1 was completed by 31 October 2022. Phase 2 has been delayed by COVID19 public health restrictions but will be completed by June 2023. Phase 3 will be completed by 31 October 2024. Phase 4 will commence in November 2023 and be completed by 31 October 2024. Analysis of data from Phases 1–3 will feed into Phase 4 to develop guidelines, COS and caregiver resources.

Our research plan has been co-designed with community partners and we have developed an integrated knowledge transfer strategy [50–52] to maximise the impact of outputs. Co-production with community-based partners, PLWD and caregivers and family will result in culturally appropriate real-world application.

## Discussion

The number of PLWD is rising significantly. It is estimated that 34% of PLWD face challenges associated with toilet use or continence [10, 11] and there is currently limited guidance for public health professionals on promotion of continence, or management of incontinence for this population. Therefore, many PLWD are experiencing reduced quality of life, and an increased likelihood of relocating into aged research care (ARC). Without appropriate assistance,

caregivers and family providing support to PLWD with toilet use and continence challenges, may face increased levels of stress, affecting both their physical and mental health. *"Incontinence is the biggest challenge facing the person we love and support, and our family. It is the issue we spend most time and resources on, and the reason for the majority of our exhaustion"* [92].

This research will translate epidemiological, behavioural and social research into guidelines for health care professionals and interventions for PLWD and caregivers and family. The study will spearhead improvements in the promotion of continence and management of incontinence for PLWD. It will enhance caregivers' access to culturally appropriate support resources and caregiving safety providing lasting societal benefits that will ultimately reduce health and social care costs. Reduced pressure on caregivers and family will lead to better quality of life (improved sleep, physical and mental health, and psychosocial outcomes such as loneliness), and delayed or reduced relocation of PLWD into ARC.

Evidence suggests that globally, PLWD experiencing difficulties with toilet use and continence are vulnerable to exclusion. The lack of instructions for health care professionals to initiate discussions about continence with PLWD, coupled with the public's reluctance to discuss the taboo subject, has the effect of marginalising and silencing this population [16]. In turn, the decreased visibility and voice of PLWD facing continence issues along with the lack of data on prevalence, does little to challenge the status quo and the scant research attention paid to the issue [16].

Health, social and cultural benefits will be achieved through the development of guidelines for clinicians and health care professionals and support resources for caregivers that will be distributed via national and community organisations. This is anticipated to facilitate improvement in the responsiveness of health and social services to meet the needs of PLWD and caregivers in relation to promoting continence and managing incontinence. Including PLWD, caregivers and family in the co-production of continence guidelines will help to avoid nihilistic clinical approaches that nothing (other than containment) can be done [16]. Culturally appropriate resources will provide practical scalable solutions for challenges associated with caregiving and promoting continence or managing incontinence for Māori and non-Māori PLWD, caregivers and family.

Academic benefits include the potential to bring about positive systems change [16]. For example, co-production methods may provide an exemplar suited to working with older people from a range of ethnicities, cultures, and in a range of settings. In the future, the research could be adapted for use with older indigenous people or ethnic minority groups in international contexts.

In order to establish the effectiveness of interventions and the co-ordinated delivery of services that meet needs during a changing illness trajectory and in multiple contexts, we need to know what/ constitutes good continence care from the perspective of PLWD, caregivers and families [16, 93].The core outcome set developed during the research will be used to assess the effectiveness of future interventions, sustaining long term benefits. To date outcomes that are important to PLWD and caregivers (but that are essential to support PLWD in the community) have rarely been measured in intervention studies. This research will help health professionals to understand key domestic issues and ensure that future interventions are relevant to PLWD, caregivers and families [94].

## Supporting information

**S1 Appendix. Focus group questions.**
(DOCX)

**S2 Appendix. Topic guide PLWD cross-sectional interview.**
(DOCX)

**S3 Appendix. Topic guide caregiver cross-sectional interview.**
(DOCX)

**S4 Appendix. Topic guide PLWD longitudinal interview.**
(DOCX)

**S5 Appendix. Topic guide caregiver longitudinal interview.**
(DOCX)

**S6 Appendix. Inclusivity in global research.**
(DOCX)

## Author Contributions

**Conceptualization:** Vanessa Burholt.

**Funding acquisition:** Vanessa Burholt, Kathryn Peri, Deborah Balmer, Gary Cheung, Rosemary Gibson, Ngaire Kerse, Anna Michele Lawrence, Tess Moeke-Maxwell, Avinesh Pillai, Lisa Ann Williams.

**Methodology:** Vanessa Burholt, Kathryn Peri, Sharon Awatere, Deborah Balmer, Gary Cheung, Rosemary Gibson, Ngaire Kerse, Anna Michele Lawrence, Tess Moeke-Maxwell, Erica Munro, Yasmin Orton, Avinesh Pillai, Arapera Riki, Lisa Ann Williams.

**Project administration:** Vanessa Burholt.

**Writing – original draft:** Vanessa Burholt, Kathryn Peri.

**Writing – review & editing:** Sharon Awatere, Deborah Balmer, Gary Cheung, Julie Daltrey, Jaime Fearn, Rosemary Gibson, Ngaire Kerse, Anna Michele Lawrence, Tess Moeke-Maxwell, Erica Munro, Yasmin Orton, Avinesh Pillai, Arapera Riki, Lisa Ann Williams.

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
