## [Decision Letter · Decision Letter 0]

12 Jun 2023

PONE-D-23-04272Improving continence management for people with dementia in the community in Aotearoa, New Zealand: Protocol for a mixed methods studyPLOS ONE

Dear Dr. Burholt,

Thank you for submitting your manuscript to PLOS ONE. After careful consideration, we feel that it has merit but does not fully meet PLOS ONE’s publication criteria as it currently stands. Therefore, we invite you to submit a revised version of the manuscript that addresses the points raised during the review process. The reviewers have outlined a series of minor revisions and points that require clarification.

We look forward to receiving your revised manuscript.

Kind regards,

Andrew Harding, PhD

Academic Editor

PLOS ONE

Journal Requirements:

2. Please describe in your methods section how capacity to provide consent was determined for the participants in this study. Please also state whether your ethics committee or IRB approved this consent procedure. If you did not assess capacity to consent please briefly outline why this was not necessary in this case.

3. Please include a complete copy of PLOS’ questionnaire on inclusivity in global research in your revised manuscript. Our policy for research in this area aims to improve transparency in the reporting of research performed outside of researchers’ own country or community. The policy applies to researchers who have travelled to a different country to conduct research, research with Indigenous populations or their lands, and research on cultural artefacts. The questionnaire can also be requested at the journal’s discretion for any other submissions, even if these conditions are not met.  Please find more information on the policy and a link to download a blank copy of the questionnaire here: https://journals.plos.org/plosone/s/best-practices-in-research-reporting. Please upload a completed version of your questionnaire as Supporting Information when you resubmit your manuscript.

Additional Editor Comments:

Thank you for submitting your manuscript. I happy to be able report that we can recommend your manuscript for publication, subject to you addressing some minor revisions and points for clarification as outlined by both reviewers.

Reviewers' comments:

Reviewer's Responses to Questions

**Comments to the Author**

1. Does the manuscript provide a valid rationale for the proposed study, with clearly identified and justified research questions?

Reviewer #1: Yes

Reviewer #2: Yes

2. Is the protocol technically sound and planned in a manner that will lead to a meaningful outcome and allow testing the stated hypotheses?

Reviewer #1: Yes

Reviewer #2: Yes

3. Is the methodology feasible and described in sufficient detail to allow the work to be replicable?

Reviewer #1: Yes

Reviewer #2: Yes

4. Have the authors described where all data underlying the findings will be made available when the study is complete?

Reviewer #1: Yes

Reviewer #2: Yes

5. Is the manuscript presented in an intelligible fashion and written in standard English?

Reviewer #1: Yes

Reviewer #2: Yes

6. Review Comments to the Author

You may also provide optional suggestions and comments to authors that they might find helpful in planning their study.

Reviewer #1: This paper describes a comprehensive programme of work in a hugely important, but under-explored area. I have a few minor comments/suggestions for the authors:

• Please expand on how the work in Ref 16 provides a theoretical framework (line 190).

• For objective 1 – please explain why this is limited to people receiving homecare. Presumably for pragmatic data availability reasons?

• In Phase 2, the analysis of the focus groups focuses on identifying shared group understanding of policy ideals & language. Will it also specifically identify perceptions of effective strategies for promoting continence & managing incontinence (Aim ii)?

• Consider using the word ‘toilet-use’ rather than ‘toileting’ – toileting is considered by some to be pejorative (although this might be cultural).

Reviewer #2: Reviewer’s report: Improving continence management for people with dementia in the community in Aotearoa, New Zealand: Protocol for a mixed methods study

This protocol paper proposes a study which seeks to examine the nature and extent of challenges and practices which are in operation in terms of continence care with the aim of improving confidence related outcomes for adults with dementia in Aotearoa, NZ.

Is this a mixed methods study? The proposal looks like a sequential qualitative – quantitative design – is there a planned synthesis of the two methods planned?

Introduction.

The sentence on line 75 seems a little redundant or out of sequence. Id omit it having prespecified the local issue

The statement at line 82-83 would benefit from a qualifying reference.

Continence services & support:

Is ref 16 specific to the proposed setting – in some jurisdictions there are systematized continence services and planned care delivery.

Reference 29 is a little old more recent research is more contradictory. There seems to be consistency around FI & dementia -perhaps update this, particularly in light of the profound change in institutionalised older adult populations and the impact of aging in place?

Are there data on incontinence on Maori attitudes, beliefs, understanding of incontinence and current management etc?

Summary

Line 145 – a generic or local statement?

Ref 46 – has also been updated – particularly in the evaluation section.

Phase 1:

Please detail the frequency of RAI-HC assessment? In our system it is yearly – and this severely limits the ability to calculate incidence

Have the accuracy of the continence variables been validated locally?

Who collects the data?

Sex, rather than gender – is gender collected in RAI-HC? Should mobility status not be included as a single variable?

Phase 1 makes no mention of how incidence will be calculated or expressed

Phase 2: this certainly “feels” like a mixed methods phase – please could the authors articulate how the quantitative policy review will aid the qual portion and how the synthesis will be approached? To what end will the product be put?

Phase 3: this combines a secondary analysis of existing data combined with primary data collection. I wasn’t sure whether the primary data will be collected form PLWD with incontinence (and their carepartners)

Have the proposed interview guides been tested for cultural appropriateness and meaning?

On line 338 – I got confused as to the nature of the quant data in this phase – other than the detailed questionnaires and how these data might be used to construct participant profiles.

Phase 4: how will the researchers ensure that the voice of the end users (PLWD / care partners Maori / non Maori) are not lost in the process? How will evidence informed interventions be promoted and preserved in the guidance?

7. PLOS authors have the option to publish the peer review history of their article (what does this mean?). If published, this will include your full peer review and any attached files.

Reviewer #1: No

Reviewer #2: No

---

## [Author Response · Author response to Decision Letter 0]

18 Jun 2023

Title page has been amended.

2. Please describe in your methods section how capacity to provide consent was determined for the participants in this study. Please also state whether your ethics committee or IRB approved this consent procedure. If you did not assess capacity to consent please briefly outline why this was not necessary in this case.

Additional information has been added a Line 444 “The researchers will ask participants to describe in their own words what the study is about, and what they are being asked to do.”

3. Please include a complete copy of PLOS’ questionnaire on inclusivity in global research in your revised manuscript. Our policy for research in this area aims to improve transparency in the reporting of research performed outside of researchers’ own country or community. The policy applies to researchers who have travelled to a different country to conduct research, research with Indigenous populations or their lands, and research on cultural artefacts. The questionnaire can also be requested at the journal’s discretion for any other submissions, even if these conditions are not met. Please find more information on the policy and a link to download a blank copy of the questionnaire here: https://journals.plos.org/plosone/s/best-practices-in-research-reporting. Please upload a completed version of your questionnaire as Supporting Information when you resubmit your manuscript.

This form has been completed, but the research is not being conducted outside of the researchers’ own community/country. All of the research team live and work in New Zealand. The Māori arm of the research is led by Māori researchers who live and work in the same community. As PI I hold a 0.2FTE position in the UK as well as 1FTE in New Zealand. I have permanent residency in New Zealand.

As PI I have a joint position at the University of Auckland (I live in New Zealand) which is my primary email address and contact address, and in Swansea University, UK (0.2FTE), both affiliations are noted on the title page. I have access to APC funding from both Universities, and in this case only Swansea University have a funding agreement with Plos One. 

Now included information on supporting information

One reference has been revised on the advice of the reviewers [46]

Reviewer #1: This paper describes a comprehensive programme of work in a hugely important, but under-explored area. I have a few minor comments/suggestions for the authors:

• Please expand on how the work in Ref 16 provides a theoretical framework (line 190). 

Additional information has been provided about the structure of the theoretical framework and how it has influenced the study: Line 204-214, Note that line numbers refer to the tracked version of the MS. 

• For objective 1 – please explain why this is limited to people receiving homecare. Presumably for pragmatic data availability reasons?

The reasons for a pragmatic utilisation of routinely collected data has been added lines 247-251. 

• In Phase 2, the analysis of the focus groups focuses on identifying shared group understanding of policy ideals & language. Will it also specifically identify perceptions of effective strategies for promoting continence & managing incontinence (Aim ii)?

Added to line 314. 

• Consider using the word ‘toilet-use’ rather than ‘toileting’ – toileting is considered by some to be pejorative (although this might be cultural).

All references to toileting have been changed to toilet use and we thank the reviewer for brining this to our attention.

Reviewer #2: Reviewer’s report: Improving continence management for people with dementia in the community in Aotearoa, New Zealand: Protocol for a mixed methods study

This protocol paper proposes a study which seeks to examine the nature and extent of challenges and practices which are in operation in terms of continence care with the aim of improving confidence related outcomes for adults with dementia in Aotearoa, NZ.

Is this a mixed methods study? The proposal looks like a sequential qualitative – quantitative design – is there a planned synthesis of the two methods planned?

This is a mixed methods study with Phase 1 drawing on quantitative methods and phases 2-3 drawing on qualitative methods. Please see line 204. Phase 4 is a transformative sequential phase during which the data are synthesised and used during the co-creation of resources, guidelines and a core outcome set - as described in Chapter 5, Researching ageing by Christin Victor et al. https://library.oapen.org/bitstream/id/ae750b1a-d62b-4ef3-99a7-7ef14da91e32/9780367507558_text.pdf)

Introduction.

The sentence on line 75 seems a little redundant or out of sequence. Id omit it having prespecified the local issue

We have not omitted this line because we also want to contextualise the study in the global context. Our methods have global relevance, given the paucity of research in this area. The protocol has relevance for replication elsewhere. 

The statement at line 82-83 would benefit from a qualifying reference.

Reference Added to line 92. 

Continence services & support:

Is ref 16 specific to the proposed setting – in some jurisdictions there are systematized continence services and planned care delivery.

This is now qualified line 111-113. “However, a review of literature published in English from countries in the upper quartile (above 47) of the Human Development Index demonstrated that continence care in the community is currently not systematised for PLWD and there is no structured care pathway for this population.”

Reference 29 is a little old more recent research is more contradictory. There seems to be consistency around FI & dementia -perhaps update this, particularly in light of the profound change in institutionalised older adult populations and the impact of aging in place?

The conclusion drawn in our expert review of the literature was that the evidence on incontinence (either UI or FI) in combination with dementia contributes to entry into residential care/nursing home. To our knowledge, since this review there have not been any dramatic changes in terms of continence service provision specifically targeted for PLWD. We have changed ‘consistently’ to ‘frequently’ (line 138). 

Are there data on incontinence on Maori attitudes, beliefs, understanding of incontinence and current management etc?

This is the first study that will address and explore values and beliefs around incontinence for PLWD. In our first article on prevalence and incidence of FI, we have identified one article with people living with stroke, that pinpoints inequities in access to health care (based on discrimination and unsafe cutural practices) as one of the experience contributing factors to the experiences of Māori. We have incorporated ‘wairua’ and ‘tikanga’ and other Māori concepts into our exploratory work in Phase 3 ( i.e. looking at spiritual beliefs and incorporating culturally safe practices into our research methods) but are also looking at the perceptions and experience of access to health care services. See lines 147 -155

Summary

Line 145 – a generic or local statement?

We have left this line as is, as the previous qualification demonstrated that this is based on international evidence. 

Ref 46 – has also been updated – particularly in the evaluation section.

The MRC guidance has been updated since we wrote our protocol for submission to the funders. We have amended the protocol and have updated the citation and added the following text, line 215-221 “Drawing on a framework for developing complex interventions [46] the study represents the first steps in the development or identification of an intervention. The 3-year project is the first stage in a pipeline of research, and considers context (Phase 1-3), programme theory (Phase 2), diverse stakeholders’ perspectives, and key uncertainties (Phases 2-4) to identify the most promising strategies and interventions for promoting continence and managing continence for PLWD living in the community (Phase 4) for future testing [46].” The notion that this is the first stage in the pipeline of research was identified earlier in the MS.

Phase 1:

Please detail the frequency of RAI-HC assessment? In our system it is yearly – and this severely limits the ability to calculate incidence

Who collects the data?

Statement about interRAI assessment frequency and assessors has been added lines 230-237.

Have the accuracy of the continence variables been validated locally?

Limitations of the interRAI HC data, and areas for further research/exploration will be discussed in the articles that arise from the prevalence and incidence studies. The validity of the interRAI instruments as a comprehensive geriatric assessment has led to its implementation in 35 countries.

Sex, rather than gender – is gender collected in RAI-HC? 

InterRAI HC forms in NZ refer to gender.

Should mobility status not be included as a single variable?

We have selected to use ADLs and physical activity in the analysis and will not be using a single mobility variable.

Phase 1 makes no mention of how incidence will be calculated or expressed

This has been added to the MS lines 260-267and Lines 278-281

Phase 2: this certainly “feels” like a mixed methods phase – please could the authors articulate how the quantitative policy review will aid the qual portion and how the synthesis will be approached? To what end will the product be put?

Phase 2 is not quantitative analysis. The text states that this is “Interpretative policy analysis [61] comprising documentary analysis of international guidelines [62] and regional publicly funded continence services policy aspirations and an analysis of staff perceptions and experiences.” 

Phase 3: this combines a secondary analysis of existing data combined with primary data collection. I wasn’t sure whether the primary data will be collected form PLWD with incontinence (and their carepartners).

Amended lines 344 to indicate this is PLWD experience difficulties with incontinence and their caregivers and family. 

Have the proposed interview guides been tested for cultural appropriateness and meaning?

The Māori topic guides have been developed by the Māori Research Fellow with support from a Māori Research Assistant (kuia, female Māori elder). This has been added to the text Line 389-391

On line 338 – I got confused as to the nature of the quant data in this phase – other than the detailed questionnaires and how these data might be used to construct participant profiles.

The reviewer has correctly identified the use of the quantititive data in Phase 3: to construct participants profiles. E.g., for PLWD to describe the sample in terms of cognitive function, gender, age, ethnicity, level of education. We have added additional words to this section “will be used to construct participant profiles characterising the samples of PLWD and their caregivers” (line 376)

Phase 4: how will the researchers ensure that the voice of the end users (PLWD / care partners Maori / non Maori) are not lost in the process? 

The MS notes that all three sub-studies in Phase 4 are co-created – the purpose of co-creation is to ensure that the voices of our participants are heard. The team are experts in co-creation (e.g. the PI is Co-Director of the Centre for Co-Created Ageing Research), and the proposal itself is based on the recommendations of an Expert Advisory Group that included PLwD and the carers. 

How will evidence informed interventions be promoted and preserved in the guidance?

This is the first step in a pipeline of research (see above). We will identify the most promising strategies, but these will be robustly tested in our next research programme. The guidance for health care professionals will be based on our policy review (including international best practice policy in comparison to NZ policy), health professionals’ views, and the experience of PLWD and their caregivers (e.g. what works, what needs to improve).

Resources for PLWD and caregivers will be co-created, and will be based on the most promising strategies that proved to be useful, and produce good outcomes for those involved in the study.

All figure files have already been checked through PACE to meet PLOS requirements.

---

## [Editor Report · Decision Letter 1]

3 Jul 2023

Improving continence management for people with dementia in the community in Aotearoa, New Zealand: Protocol for a mixed methods study

PONE-D-23-04272R1

Dear Dr. Burholt,

We’re pleased to inform you that your manuscript has been judged scientifically suitable for publication and will be formally accepted for publication once it meets all outstanding technical requirements.

Kind regards,

Andrew Harding, PhD

Academic Editor

PLOS ONE

Additional Editor Comments (optional):

Thank you for taking the time to revise and resubmit your manuscript based on the reviews. After reviewing your response to reviewer comments, and the revised manuscript, I am happy to recommend your manuscript for publication. This is important and robust research, and I look forward to seeing the findings once they are published.
---

## [Editor Report · Acceptance letter]

6 Jul 2023

PONE-D-23-04272R1 

Improving continence management for people with dementia in the community in Aotearoa, New Zealand: Protocol for a mixed methods study 

Dear Dr. Burholt:

I'm pleased to inform you that your manuscript has been deemed suitable for publication in PLOS ONE. Congratulations! Your manuscript is now with our production department. 

Kind regards, 

on behalf of

Dr. Andrew Harding 

Academic Editor

PLOS ONE